



# Composite Catalogues of Optical and Fluorescent Signatures Distinguish Bioaerosol Classes

**Authors & Institutional Affiliations**
First Author:          M. Hernandez*
Department of Civil, Environmental and Architectural Engineering,
UCB 428
University of Colorado
Boulder, CO 80309
USA
mark.hernandez@colorado.edu
Second Author:         A. Perring
National Oceanic and Atmospheric Administration
325 Broadway
Boulder, CO 80305
USA
Cooperative Institute for Research in Environmental Sciences
University of Colorado
Boulder, CO 80309
USA
Third Author           K. McCabe
Department of Sciences
Columbia George Community College
400 East Scenic Drive
The Dalles, OR 97058
USA
Fourth Author:         G. Kok
Droplet Measurement Technologies
2545 Central Ave
Boulder, CO 80301
USA
Fifth Author:          G. Granger
40                         Droplet Measurement Technologies
Sixth Author:          D. Baumgardner
43                         Droplet Measurement Technologies
* Corresponding Author



**ABSTRACT**
Rapid bioaerosol characterization has immediate applications in the military, environmental and public
health sectors. Recent technological advances have facilitated single-particle detection of fluorescent
aerosol in near real-time; this leverages controlled exposures with single or multiple ultraviolet
wavelengths, followed by the characterization of associated fluorescence. This type of Ultraviolet
induced fluorescence has been used to detect some intact airborne microorganisms in laboratory studies,
and has been extended to field studies which implicate bioaerosol to compose a substantial fraction of
supermicron atmospheric particles. To enhance the information yield which new generation fluorescence
instruments provide, we report the compilation of a systematic referential catalogue including more than
fifty pure cultures of common airborne bacteria, fungi and pollens. This catalogue juxtaposes intrinsic
optical properties and multiple bandwidths of fluorescence spectra, which manifest to clearly distinguish
between major classes of airborne microbes and pollen.
**KEYWORDS:**
Bioaerosol; Fluorescence; Aerosol Cytometry



## INTRODUCTION

Fluorescence aerosol interrogation is gaining increased attention for its ability to characterize particulate matter suspended in both indoor and outdoor environments (Huffman et al., 2010; Sivaprakasam et al., 2011). When simultaneously reporting across multiple bandwidths, ultraviolet induced fluorescence (UVIF) has successfully detected airborne microbes in bench-scale chamber studies (Healy et al., 2012; Kaye et al., 2005; Toprak and Schnaiter, 2013) and been applied to support large-scale aerosol monitoring campaigns where fluorescent particle cohorts indicated significant bioaerosol contributions to airborne organic carbon pools (Gabey et al., 2010; Gabey et al., 2013; Perring et al., 2015; Poschl et al., 2010).

A variety of biogenic fluorophores have evolved in microorganisms and pollen grains, many of which are relevant to UVIF characterization of aerosols (Lakowicz, 2006; Poehlker et al., 2012). These flourophores include metabolic mediators (e.g. NADH) that are widely conserved throughout the microbial and plant worlds. Other fluorescently active biopolymers include a wide variety of structural proteins and pigments; however these microbial compounds are tremendously variable in conformation and intracellular quantity (Hill et al., 2014; Kepner and Pratt, 1994; Madigan, 2011). As such, the distribution of particle fluorescence yields from some microbial bioaerosols has been reported to have sensitivities to age (Kanaani et al., 2007) as well as cultivation history and environmental conditions (Saari et al., 2014; Uk Lee et al., 2010).

Sensitivity analyses of bioaerosol spectra from different UVIF instrument configurations are emerging in the literature (Agranovski et al., 2003; Huffman et al., 2010; Poehlker et al., 2012), as are generalized performance indices designated as apparatus‒specific detection efficiencies (Saari et al., 2014). These UVIF metrics include, but are not limited to, photomultiplier thresholds, fluorescent particle fraction (FPF) recoveries, and specific quantum yield comparisons. The calibration of modern UVIF instruments has been predominantly executed with thermoplastic spheres that are impregnated with colloidal metals or artificial fluorophores. In this context, conventional UVIF calibrations have relatively short stability windows of specific intensity which are useful for bioaerosol standardization, because of their sensitivity to temperature and light. And while these calibrants are well characterized from a chemical perspective, they are poor biophysical analogues to the bioaerosols they are meant to reference.

There is increasing interest in the utility of UVIF for environmental and autecological studies (Despres et al., 2012), yet there remains no unified approach for calibrating and cataloguing the optical signatures associated with primary biological airborne particles (PBAPs). While many PBAPs fluoresce, interpreting UVIF measurements for bioaerosol characterization presents challenges that can only be





addressed using referenced distributions of fluorescence emissions complemented with conventional
optical properties. In response, we present a laboratory characterization of aerosolized pure cultures of
common airborne bacteria (14), fungi (29) and pollen grains (12) using a portable Wideband-Integrated
Bioaerosol Sensor ((WIBS), Droplet Measurement Technologies, Boulder, CO, USA). We report a
systematic compilation of optical and fluorescence properties that can be reproduced and expanded as a
bioaerosol reference basis for new generations of UVIF instrumentation.
**MATERIALS & METHODS**
A cohort of conventional and fluorescent bioaerosol spectra were obtained from consecutively
aerosolizing pure cultures of bacteria, fungi and pollens in an environmentally controlled chamber. The
chamber into which all bioaerosol cultures were introduced was cubic, and constructed of grounded 1.25
cm (thick) Lucite; chamber air was continuously mixed by two 1.5W fans (**Fig. 1**). Comprehensive
details of chamber construction and its operation for pure culture bioaerosol characterization have been
previously described (Peccia et al., 2000). A WIBS continuously drew chamber air at 1.2 L/min through
0.25m of conductive neoprene tubing for these tests. The chamber was vented to a hood at ambient
pressure (c.a. 88 kPa); RH was held between 25 and 40%; temperature was held between 20-22C; and,
bioaerosol was allowed to equilibrate to these conditions for 5 minutes prior to UVIF sampling.

**Bacteria.** Fifteen pure bacterial cultures were grown into early stationary phase using standard protocols
described by the American Type and German Culture Collections (ATCC and DSMZ). These cultures,
listed with their accompanying fluorescence distributions in **Fig. 2**, have been used as models for the
environmental behavior of airborne bacteria with public health relevance (CDC, 2013). Upon entering
stationary phase, they were washed by sequential centrifugations in cold, ultrapure deionized water, and
immediately aerosolized into a chamber using a six-jet Collison nebulizer. Direct microscopy showed
these bacteria remained intact after inspecting nebulizer reflux immediately following their aerosolization.
**Fungi**. Twenty-nine pure cultures of commonly occurring indoor fungi (Vesper et al., 2007) were
cultured until they presented an obvious spore-bearing physiology after being inoculated on MEA agar
and held at 20C and 25% RH. As judged by direct microscopy, copious fungal spores were presented
between 18 and 28 days, depending on the species. These fungal cultures, also presented in **Fig. 2,** were
dry aerosolized directly off their host agar into the chamber using dry ultrapure nitrogen guided through
sterile glass containments specifically designed for this purpose (modified fungal spore source strength



tester (FSSST)(Sivasubramani et al., 2004)).  Direct microscopy showed these fungal spores dominated
the airborne biomass with less than 1% of the airborne volume as associated hyphae.
**Pollen.**  Thirteen pure stocks of common temperate tree and grass pollens (O'Connor et al., 2013) were
obtained from a collection at the Denver Botanic Garden.  These pollen stocks, also listed in **Fig. 2,** were
aerosolized into the chamber with the same FSSST configuration used for fungi, with notable
fractionation of some grains as judged by microscopy.
**Optical spectra acquisition.**  The WIBS (v 4.0), previously described by Healy and coworkers (2012)
uses dual-wavelength excitation and fluorescence detection, while simultaneously measuring
characteristic dimensions from scattered light.  With the portable WIBS variant used here, fluorescence
was induced by sequential exposure to UV irradiation from a flashlamp filtered at 280 nm and 370 nm.
Fluorescence emitted due to 280 nm excitation was detected in two wavebands, 310-400 nm, and 420-
650 nm, using dedicated photomultipliers.  Fluorescence emitted due to 370 nm excitation was detected
between 420-650 nm.  Optical diameter was determined by light scattered from exposure to a 635nm
laser; it is reported here as an equivalent optical diameter (EOD), defined as the diameter of a spherical
particle, with a fixed refractive index (relative to that calibrated with 2.0 and 2.8 μm latex beads in air
between 25-45% RH), scattering the same light intensity as the measured bioaerosol.  These optical
properties were recovered from a minimum of between $10^3$ and $10^4$ bioaerosol particles for each pure
culture aerosolized over at least a 5 min period.  Between each aerosolization challenge, the chamber
was evacuated using a high volume HEPA filter such that total particle counts were below $10^2$ m$^{-3}$, and
the chamber subsequently purged with ethanol vapor and/or ozone while illuminated with UV light.

Following the annotation introduced by Perring and coworkers (2015), particle fluorescence was
categorized as one of seven types, which considers each of three fluorescence bandwidths individually,
as well as in all possible combinations.  Here the subscripts denote excitation wavelengths, and
parentheticals indicate the emission bandwidths observed:


151                    TYPE $\mathbf{A_{280}}$ =(310-420);    TYPE $\mathbf{B_{280}}$=(420-650);    TYPE $\mathbf{C_{370}}$ = (420-650)


Leveraging these measured quantities, the following metrics were analyzed and compiled for each
bioaerosol: (i) the frequency of particles that could be segregated by fluorescence signal (bandwidth) into





any of the seven types; (ii) the average fluorescence intensity within each bandwidth (if any); and (iii) the
average EOD of each particle type.
To assess the potential effect of aging, fluorescence emissions from airborne spores of *Phoma spp.*,
*Penicillum chrysogenum*, *Cladosporium cladosporides*, and *Aspergillus versicolor* that were less than 28
days old, were compared to the spectra of spores from these same cultures, which were allowed to age
180 days (raised and maintained under identical environmental conditions).
**RESULTS & DISCUSSION**
**Composite optical recognition patterns of bioaerosol classes:** The seven fluorescence categories,
the fluorescence intensities and the relative size expressed by the EOD provide a
multidimensional matrix from which composite signatures can be constructed for the three
bioaerosol classes tested here. Figure 2 shows the EOD and the distribution of fluorescence type for
each of the cultures studied while Figure 3 shows the normalized fluorescence intensity they emitted. The
pure culture bioaerosols used to challenge this WIBS distinctly clustered into respective physiologic
groups. All of the bacterial cultures aerosolized, clustered below an EOD of 1.5 µm while presenting the
weakest fluorescence intensities of the bioaerosols observed. With the exception of spore forming
*Bacillus subtilis*, bacterial bioaerosol was dominated by a single fluorescence type (A). The fungal spores
aerosolized were in a markedly higher range with respect to their equivalent optical diameters (2-9 µm),
but presented several fluorescence types: A, AB, BC and ABC, with a prevalence of A and AB. While
the pollens aerosolized were overlapping in equivalent optical diameter ranges with some of the fungal
spores observed, they presented significantly different fluorescence type distributions, which were
dominated by combinations of fluorescent types BC and ABC. We note that many of the pollen EODs
presented here are considerably smaller than the true diameter of their nascent intact-grain size, which we
attribute to pollen fragmentation during collection, storage or aerosolization. Observational studies
indicate that pollen grain fragmentation happens in the atmosphere as well (Miguel et al., 2006; Taylor et
al., 2007). The pollens also clustered based on their markedly higher fluorescence intensities relative the
other bioaerosol classes observed (**Fig. 3**).
**Fluorescence response to spore aging:** Fluorescence spectra from four pairs of young and aged fungal
spores were compared by these analyses (aged 28 vs. 180 days). Of these, two presented no discernable
change in the EOD and fluorescence signals used as cataloguing metrics here: *Cladosporium*
*cladosporiodes* and *Aspergillis niger*. There was however, a significant shift in the fluorescence



distribution, from younger to older spores, of the species *Pennicillium chrysogenum* and *Phoma*
*herbarium* (**Fig. 2** (**bottom**)).   More than 90% of the spores of these species presented their fluorescence
as Type A when aerosolized at an age of 28 days, but this decreased to near 70% after 180 days of culture
aging under identical conditions; the balance becoming type AB.
**Instrumental variability and gain considerations:** We present here the response of a single WIBS
instrument to pure culture bioaerosols grown, collected and aerosolized under carefully controlled
conditions. The instrument manufacturer (Droplet Measurement Technologies) typically sets detector
gains based on photomultiplier fluorescence detection from monodispursed challenges with commercially
available microspheres (polystyrene latex (PSL)), the intensity of which is known to vary from batch to
batch.  The fluorescence of PSLs however, significantly degrades on the timescale of months (even with
proper storage).  While this strategy provides general consistency across different instruments, the lack of
an absolute calibration means that there can be appreciable variability in fluorescence signal recovery of a
particular bioaerosol between one UVIF instrument and another.  In response, we aerosolized a subset of
test bioaerosol cultures to challenge two WIBS in parallel: one owned by the manufacturer, which was
used for the larger set of measurements presented here, and another owned by the Chemical Sciences
Division of the National Oceanic and Atmospheric Sciences Administration (NOAA). We find
differences in spectral classification (Figure 4), likely attributable to differing gain settings, with the
NOAA WIBS recording a larger fraction of a given particle population as having signal above threshold
in the B and C channels (both of which are detected using the same photomultiplier), than does the
manufacture's market issue WIBS.
**Implications for real-time detection of atmospheric bioaerosol:** Given the potential variability
introduced with culture age and between instruments, genera- or species-type classifications for different
bioaerosols should be treated not as absolute "signatures" but, rather, as likely groupings that, when
considered in conjunction with EOD, allow for discrimination between broad bioaerosol categories. There
is also particle-to-particle variability in the type manifestation of a given sample (i.e. not all particles of a
given species present as the same spectral type) that makes reliable identification on a single particle basis
unlikely and requires, instead, a statistical treatment.  For example, bacterial populations tend to present
as mostly type A at small (<1 μm) EOD.  Fungi present a mixture of type A, AB and ABC fluorescence at
larger EOD sizes (2 - 9μm), although the exact distribution between the dominant types may differ
substantially from that presented here depending on instrumental parameters, culture conditions and
environmental (aerosol) conditions. Finally, pollen tends to present as a mixture of C, BC and ABC and is
the only class of bioaerosol for which types C or BC are significant contributors. This facilitates



discrimination between fungi and pollen despite the fact that the likelihood of pollen fragmentation
reduces the utility of EOD as an identifier. Another distinguishing feature of pollens is their relatively
high fluorescence intensity although we note that very little is known about the temporal evolution of
fluorescent intensity over atmospherically relevant timescales of photochemical processing.
Type A was dominant in all bacterial and most fungal populations aerosolized in this study (at least for
certain gain settings, like those in the DMT WIBS) and thus EOD is an important consideration in
distinguishing bacteria from fungi. In this work, under laboratory conditions, we typically sampled single,
intact, bacterial cells. Atmospheric observations of bacteria and DNA in aerosol, however, indicate that
bacteria are frequently found associated with particle sizes larger than that of a single cell. (Burrows et al.,
2009; Shaffer and Lighthart, 1997; Tong and Lighthart, 2000; Wang et al., 2008). It is hypothesized that
this is due to the presence of either multi-cell bacterial agglomerates or particle mixtures including
bacteria and dust or leaf fragments (Bovallius et al., 1978; Lighthart, 1997) and we acknowledge that such
agglomerates could be easily confused with intact fungal spores given their spectral similarities as
detected by the WIBS. Another notable observation of the ensemble of experiments presented here, is the
lack of an appreciable fraction of type B particles associated with these pure culture bioaerosol
challenges. Type B particles are detected as a minor fraction (up to 15%) in only a few fungal and pollen
species and, therefore, any atmospheric sampling in which type B is a dominant fluorescent type should
be examined carefully for potential (non-biological) interferents. Type AC was also strikingly absent from
the measurements presented here.

**CONCLUSIONS**
Recent advances in UVIF instrumentation have led to a new generation of atmospheric particle sensors
with promising utility for rapid characterization of fluorescent bioaerosols, both indoors and out. Here we
have demonstrated the ability of this technique to characterize whole, airborne microbial cells in a
sensitive cytometric capacity.
Where conventional light scattering analyses is coupled with the spectral analyses of fluorescence
emission(s), a novel and powerful combination of phenotypic information becomes available on a high-
throughput, single-particle basis—the broader the excitation and emission spectra observed, the higher the
potential for the fundamental characterization of (bio)aerosol *in-situ*. As has been previously described,
the WIBS technology employed here demonstrates the potential of multi-channel fluorescence analyses in
a laboratory setting; however, this work describes a novel approach for compiling a primary optical



biological particle catalogue through a simple, unified reference method created under defined conditions.
Such a library can be reproduced, expanded and shared by users for inter-laboratory comparisons and/or
associations to field observations.
In the course of this analysis, the composite evaluation of optical diameter and fluorescent spectra
associated with airborne fungal spores, pollens, bacteria revealed that these primary physiologies can be
unambiguously differentiated from each other when airborne under defined conditions (culture condition,
RH and Temperature).  While deeper cluster analyses were beyond the scope of this demonstrative report,
within the fungi and pollens there emerged optically defined phenotypes, which also formed distinct sub-
groupings—an exciting cataloguing outcome with implications for future algorithmic developments and
the potential for confident identification of "interferences" which may complicate bioaerosol quantitation
in the field (Despres et al., 2012; O'Connor et al., 2013; Poehlker et al., 2012).
The identity of fluorescent aerosol, and the fraction thereof composed of primary biological materials
remains unknown in many different environments (Despres et al., 2012; Heald and Spracklen, 2009). To
this end, these findings suggest that a composite analyses of wide-band UVIF, as referenced to robust
bioaerosol libraries, offers a promising approach to better characterize airborne particulate matter in the
laboratory as well as during wide area environmental surveillance.



APPENDICES
A table containing discrete EOD and intrinsic fluorescence values observed for each culture
aerosolized (used to construct Fig. 2 and 3)

**TABLE. 1 Optical Statistical Summary of Airborne Bacteria, Fungi and Pollens Observed**

| Figure 1 Position (#), *Genus species* | Samples | EOD | Intensity | Fluorescence Type Frequency | | | | | | |
|---|---|---|---|---|---|---|---|---|---|---|
| | | | | A | B | C | AB | AC | BC | ABC |
| **Bacteria** | | | | | | | | | | |
| (1)  *Bacillius subtilis* | 19786 | 0.7 | 656 | 11 | 2 | | 87 | 0 | 0 | 0 |
| (2)  *Vibrio fisherii* | 15073 | 0.7 | 581 | 97 | 0 | 0 | 1 | 2 | 0 | 0 |
| (3)  *Bordetella pertussis* | 12834 | 0.5 | 84 | 93 | 0 | 1 | 1 | 3 | 0 | 1 |
| (4)  *Psuedomonas aurigenosa* | 28190 | 0.4 | 248 | 94 | 0 | 0 | 1 | 3 | 0 | 2 |
| (5)  *Acinetobacter baumanii* | 30006 | 1.1 | 556 | 96 | 0 | 0 | 0 | 3 | 0 | 1 |
| (6)  *Micrococcus luteus* | 28197 | 0.9 | 502 | 97 | 0 | 0 | 1 | 2 | 0 | 0 |
| (7)  *Mycobacterium parafortuitum* | 18197 | 0.4 | 265 | 90 | 0 | 0 | 8 | 0 | 0 | 2 |
| (8)  *Staphylococcus aureus* | 30015 | 0.7 | 283 | 96 | 0 | 0 | 2 | 1 | 0 | 1 |
| (9)  *Burkholderia cepacia* | 30003 | 0.5 | 399 | 97 | 0 | 0 | 2 | 1 | 0 | 0 |
| (10) *Psuedomona GFP aurigenosa* | 16015 | 0.6 | 112 | 97 | 0 | 0 | 1 | 1 | 0 | 1 |
| (11) *Serratia marcesens* | 30027 | 0.6 | 393 | 96 | 0 | 0 | 3 | 1 | 0 | 0 |
| (12) *Enterobacter faceialis* | 17401 | 0.8 | 327 | 96 | 0 | 0 | 1 | 1 | 0 | 2 |
| | | | | | | | | | | |
| **Fungi** | | | | | | | | | | |
| (28) *Aspergillus niger (28 d)* | 30772 | 4.4 | 157 | 79 | 2 | 7 | 4 | 2 | 2 | 5 |
| (29) *Aspergillus niger (180 d)* | 23086 | 4.4 | 181 | 78 | 2 | 5 | 5 | 2 | 2 | 5 |
| (30) *Aspergillis versicolor* | 32620 | 4 | 304 | 67 | 2 | 0 | 9 | 0 | 8 | 14 |
| (31) *Aspergillis flavus.* | 15847 | 4 | 364 | 87 | 0 | 0 | 3 | 1 | 2 | 4 |
| (32) *Aspergillus pinenceus* | 33085 | 3.6 | 206 | 90 | 0 | 0 | 5 | 0 | 0 | 3 |
| (33) *Aspergillus syndowii* | 5098 | 3.5 | 515 | 90 | 0 | 0 | 5 | 0 | 0 | 3 |
| (34) *Aspergillus tubinigenisis* | 28801 | 2.5 | 172 | 41 | 7 | 7 | 25 | 0 | 5 | 12 |
| (35) *Botretious spp.* | 7024 | 8 | 294 | 21 | 6 | 0 | 55 | 0 | 0 | 16 |
| (36) *Cheatomium* | 19264 | 4.9 | 258 | 77 | 0 | 0 | 16 | 0 | 1 | 3 |
| (37) *Cladosporium spp* | 39294 | 3.3 | 710 | 80 | 0 | 0 | 16 | 0 | 0 | 2 |
| (38) *Cladosporium cladosporiodes (28 d)* | 26474 | 3 | 570 | 72 | 0 | 0 | 24 | 0 | 0 | 3 |
| (39) *Cladosporium cladosporiodes (180 d)* | 34085 | 3.3 | 702 | 70 | 0 | 0 | 23 | 0 | 0 | 4 |
| (40) *Cladosporium herbarium* | 2962 | 3.2 | 651 | 80 | 0 | 0 | 12 | 0 | 0 | 6 |
| (41) *Fusarium spp* | 1188 | 3.3 | 638 | 81 | 0 | 0 | 12 | 0 | 0 | 5 |
| (42) *Paecilmyces variotii* | 11188 | 4.2 | 772 | 0 | 8 | 0 | 74 | 0 | 3 | 13 |
| (43) *Pennicillium chrysogenum  (28 d)* | 14577 | 3.5 | 867 | 92 | 0 | 0 | 4 | 0 | 0 | 2 |
| (44) *Pennicillium chrysogenum  (180 d)* | 33452 | 3.2 | 733 | 73 | 0 | 0 | 23 | 0 | 0 | 2 |
| (45) *Pennicillium canescens* | 7840 | 3.1 | 659 | 90 | 0 | 0 | 5 | 0 | 0 | 4 |
| (46) *Pennicillium citreonigrum* | 8470 | 2.8 | 596 | 68 | 0 | 0 | 27 | 0 | 0 | 4 |
| (47) *Pennicillium commune* | 9255 | 3.3 | 762 | 79 | 0 | 0 | 3 | 2 | 0 | 14 |
| (48) *Pennicillium corylophium* | 12654 | 2.6 | 687 | 90 | 0 | 0 | 6 | 0 | 0 | 3 |
| (49) *Pennicillium cecumbens* | 13676 | 2.4 | 420 | 89 | 0 | 0 | 8 | 0 | 0 | 2 |





| | | | | | | | | | | |
|---|---|---|---|---|---|---|---|---|---|---|
| (50) *Pennicillium thomii* | 5739 | 3.1 | 721 | 91 | 0 | 0 | 4 | 0 | 0 | 3 |
| (51) *Phoma herbarium (28 days)* | 5518 | 2.9 | 607 | 91 | 0 | 0 | 5 | 0 | 0 | 2 |
| (52) *Phoma herbarium (180 days)* | 2648 | 3.5 | 688 | 79 | 0 | 0 | 14 | 0 | 0 | 5 |
| (53) *Stachybotrius spp.* | 12697 | 2.8 | 137 | 65 | 7 | 2 | 14 | 1 | 1 | 6 |
| (54) *Syncephalstrum racemosom* | 6753 | 2.8 | 618 | 75 | 0 | 0 | 12 | 0 | 0 | 10 |
| (55) *Triterachium spp.`* | 5595 | 2.2 | 227 | 95 | 0 | 0 | 3 | 0 | 0 | 0 |
| (56) *Ulocladium* | 4513 | 3.8 | 767 | 80 | 0 | 0 | 15 | 0 | 0 | 4 |
| | | | | | | | | | | |
| **Pollens** | | | | | | | | | | |
| (15) *Acer saccharum* | 800 | 3.3 | 753 | 6 | 1 | 3 | 1 | 1 | 17 | 68 |
| (16) *Alnus Rubra and Rugosa spp.mix* | 389 | 4.7 | 387 | 6 | 4 | 11 | 1 | 0 | 28 | 46 |
| (17) *Ambrosia trifida* | 1219 | 3.2 | 876 | 11 | 2 | 2 | 6 | 1 | 12 | 62 |
| (18) *Artemesia tridentata* | 1703 | 2.5 | 666 | 6 | 2 | 2 | 3 | 0 | 19 | 66 |
| (19) *Betula Lenta, Nigra & Populifolia* | 2208 | 2.5 | 990 | 4 | 1 | 3 | 2 | 0 | 54 | 33 |
| (20) *Carya lacinosa* | 1942 | 1.3 | 1043 | 5 | 0 | 5 | 2 | 0 | 53 | 33 |
| (21) *Eucalyptus* | 1095 | 6.1 | 481 | 2 | 1 | 13 | 0 | 0 | 47 | 34 |
| (22) *Fern (unknown source)* | 1571 | 6.8 | 373 | 1 | 13 | 17 | 1 | 0 | 38 | 28 |
| (23) *Fragus americana* | 1991 | 1.8 | 1155 | 6 | 2 | 6 | 3 | 0 | 60 | 20 |
| (24) *Juglands nigra* | 1968 | 1.2 | 1197 | 3 | 1 | 4 | 1 | 0 | 62 | 26 |
| (25) *Juniper ashei* | 2536 | 3 | 766 | 0 | 1 | 1 | 0 | 0 | 16 | 79 |
| (26) *Morus rubra* | 4232 | 2.8 | 1229 | 0 | 0 | 0 | 0 | 0 | 1 | 96 |
| (27) *Phelum pratense* | 3390 | 1.5 | 936 | 2 | 0 | 1 | 0 | 0 | 11 | 82 |




## ACKNOWLEDGMENTS

This work is the result of a collaboration of senior instrument developers, agency researchers and academics in the form of an interagency, university-industry cooperative. It was funded in part by a grant from the National Science Foundation, Division of Bioenvironmental Sciences, (BES 1134594) and by in-kind contributions from Droplet Measurement Technologies, Research and Development group. Salary support for AEP was from the NOAA Atmospheric Composition and Climate Program and the NOAA Health of the Atmosphere Program.




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



**TABLES AND FIGURES**

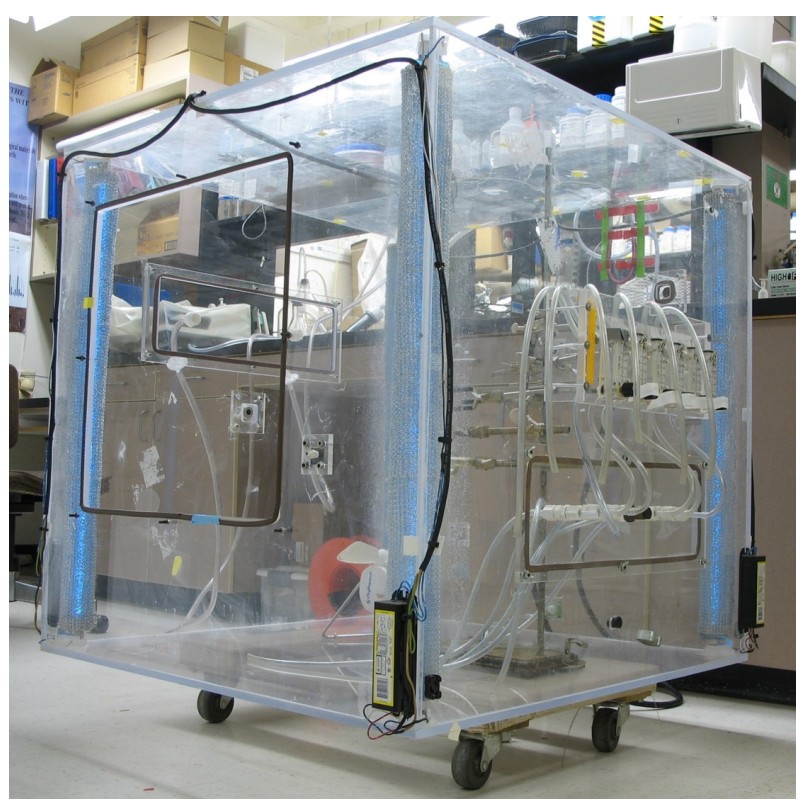


**Figure 1.** Photograph of bioaerosol test chamber. In independent trials, dry ultrapure nitrogen aerosolized and
diluted different pure cultures of bacteria, fungi and pollens into a temperature (20C) and RH (25-40%) controlled
chamber (890L). Bioaerosol delivery vessels were fitted with steel ports, and all gas streams were contained in 0.4
cm conductive neoprene tubing, carrying 151kPa at flow rate of 12 l/min into the chamber.



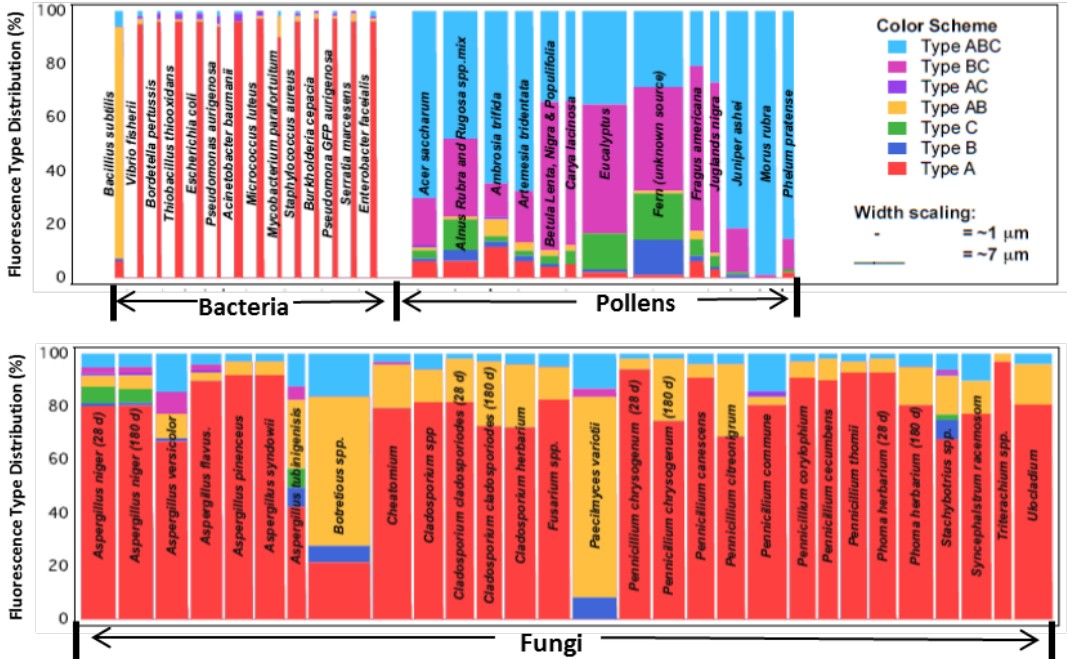

**Figure 2.** Juxtaposition of optical and fluorescence properties of aerosolized pure cultures of bacteria and pollens (**top**), and fungal spores (**bottom**). Fluorescence type distribution is defined by excitation and emission from any of three possible channels alone (A, B or C), or in any combination. Bar width is proportional to equivalent optical diameter.

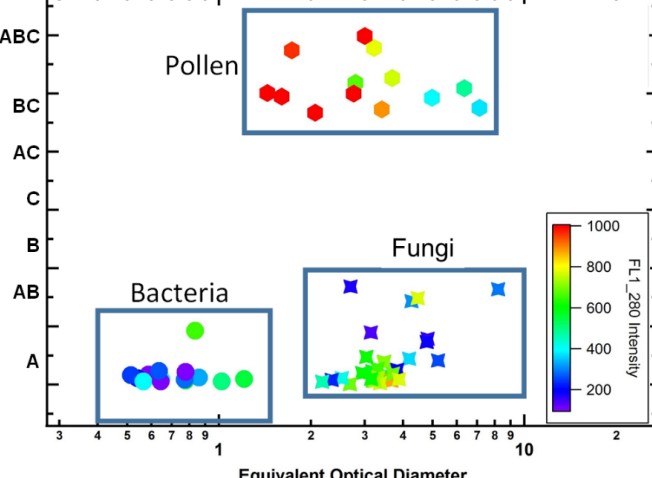

**Figure 3.** Juxtaposition of optical and fluorescence properties of aerosolized pure cultures of bacteria (●), fungi spores (**X**), and pollen grains (♦). Fluorescence type is the mode associated with each culture aerosolized. Fluorescence intensity is a relative scale.




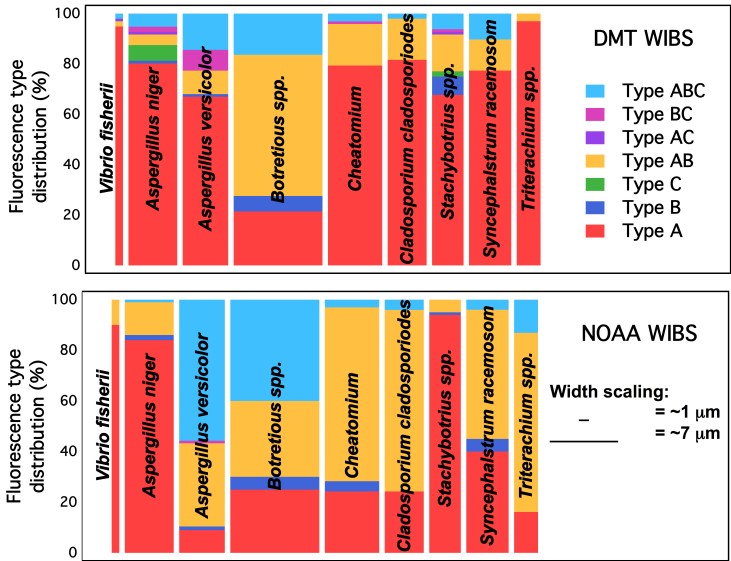


Figure 4. Comparison of type classifications for two different WIBS instruments over a subset of the pure
bioaerosol cultures used in this analysis. The top panel shows the type classification breakdown in the DMT WIBS
which was used for all of the experimental trials presented here. The bottom panel shows the type classification
breakdown in the NOAA WIBS, used in parallel for a subset of the species used in the larger work.