# Peer review of "Composite Catalogues of Optical and Fluorescent Signatures Distinguish Bioaerosol Classes"

_Atmospheric Measurement Techniques, 2015_

## Referee Comment (RC1) · Anonymous Referee #2 · 8 Feb 2016

Summary:

The manuscript by Hernandez et al. entitled "Composite Catalogues of Optical and Fluorescent Signatures Distinguish Bioaerosol Classes" presents fluorescence data of a broad selection of standard bacteria, fungal spores, and pollen from laboratory measurements. Their work intends to establish a data base by compiling the fluorescence signatures from various microorganisms based on a unified experimental procedure. The authors claim that this library can serve as a reference basis in future (ambient) bioaerosol studies, using autofluorescence based instruments (i.e., the WIBS).

One of the major challenges in autofluorescence based bioaerosol analysis is the diversity of the airborne organisms as well as the diversity of different fluorescent molecules inside these cells. Therefore, reference fluorescence data/spectra are needed to clas-

sify and interpret the output from online fluorescence detectors (e.g., the WIBS). Accordingly, the overall aim of this study is a very useful one.

However, I think the manuscript suffers from a number of major issues that require more attention. I have listed my major concerns below. Overall, I recommend that the manuscript is appropriate for AMT and will probably receive a lot of attention from the bioaerosol community. However, the manuscript in its current form needs a major revision.

General Comments:

1) It seems that the manuscript was written in a rush. There is a rather high density of typos (some are collected below under minor comments). Moreover, the discussion of the results is rather short and ignores most of the previous studies that have reported related results (see major comment 5 below). A striking example for the improvable quality is the fact that about half of the Latin names for the reference species (a focal point of this study) are misspelled. I am just giving selected examples: "bacillius subtilis" vs. "bacillus subtilis" (#1 in Table 1); "psuedomona aurigenosa" vs. "pseudomonas aeruginosa" (#10 in Table 1); "pennicillium" vs. "penicillium" (#43-49 in Table1); "artemesia" vs. "artemisia" (#18 in Table 1); "fragus" vs. "fagus" (#23 in Table 1).

2) In lines 97-99, the authors express the main objective of the study: "We report a systematic compilation of optical and fluorescence properties, which can be reproduced and expanded as a bioaerosol reference basis for new generations of UVIF instrumentation". I have some objections here.

- First, if the collection of reference organisms is called "systematic", some relevant studies should be cited to underline that the selection of organisms can indeed serve as representatives for the atmospheric bioaerosol population.

- Second, the authors claim that the results can be "reproduced" after stating in line

79 that fluorescence properties strongly depend on the "cultivation history" of the organisms. In its current state, the experimental section is rather vague in terms of "cultivation history". The following basic information is missing: (i) strain designation for the ATCC and DSMZ cultures, (ii) the basic cultivation protocols for fungi and bacteria, (iii) cultivation times should be specified more precisely, (iv) the term "obvious spore-bearing physiology" should be explained more clearly, (v) the collection procedure and age of the pollen samples should be mentioned.

- Third, I do not really agree that the reported fluorescence data can serve as a "reference basis for new generations of UVIF instrumentation". My feeling is that this statement is too broad. Given that the "new generations of UVIF instrumentation" have different excitation and/or emission specifications, I wonder if the reported data set, which is defined by the WIBS optical design, is still useful as a reference. I think that this is rather a "reference basis" for future WIBS measurements.

3) In lines 128 and 129, the authors state that pollen samples were aerosolized "with notable fractionation of some grains". That is a major aspect and important bias. What percentage of pollen grains is "fractionated"? What does it imply for the reported results? The pollen sizing deviates strongly. The equivalent optical diameters (EOD) that have been obtained for all pollen samples in this study are below 7 $\mu$m. The physical diameter of most pollen grains is > 10 $\mu$m (Despres et al., 2012). For some of the reported species the EOD in this study shows substantial deviation from (optical and physical) diameters in previous reports. For example (i) artemisia tridentata: 3.2 vs. ~21 $\mu$m; (ii) betula: 2.5 vs. ~24 $\mu$m; (iii) juglans nigra: 1.2 vs. 37 $\mu$m; (iv) phleum pratense: 1.5 vs. 34 $\mu$m (Healy et al., 2012a;Pohlker et al., 2013). The strong deviation suggests that probably only pollen fragments have been sampled. This strong deviation and the influence of the "fractionation" clearly need a careful discussion. In lines 175-180, the authors provide a brief explanation of the fragmentation. The atmospheric fragmentation of pollen grains is a phenomenon that is by far not fully understood yet. Keeping this in mind, I am not sure the presented pollen samples can serve as a "reproducible" "reference" compound in this study. Regarding this major bias, it is maybe better to skip the pollen samples in the present study.

4) Line 116: Any comment on the ultra-pure water and the Collison nebulizer, which both are pretty harsh treatments? Do you expect that the bacteria are alive or killed afterwards? Specify the type of microscopy that has been used. Does it allow to confirm cell integrity for cells < 1 $\mu$m?

5) This study presents the fluorescence properties from a variety of different organisms. However, most of the scientifically interesting aspects are not discussed in the results and discussion section. Moreover, several previous studies have reported fluorescence properties from bacteria, fungal spores, and pollen already. None of them is cited and used in the discussion to examine if the reported results agree with what is known. I understand that this is a technical paper. Moreover, I appreciate if papers are kept short and concise. However, a certain extent of discussion is still desirable in technical papers to put the results in the context of the existing knowledge. I think that the following aspects require (at least some) more explanation/discussion:

- In lines 162-176, the authors summarize the observed properties of the major classes (bacteria, fungal spores, pollen). The intensities of the observed fluorescence are lowest for the bacteria and highest for the pollen grains and, thus, correlate with size. It is well known that bioaerosol fluorescence intensity strongly depends on particle size (Hill et al., 2001;Hill et al., 2015;Healy et al., 2012b). I think that the fluorescence-size dependence is important information to explain the reported results. Cite some previous studies here.

- In line 169-170, the authors state that "with the exception of spore forming bacillus subtilis, bacterial bioaerosol was limited to a single fluorescence type (A)". Does this sentence imply that the formation of spores is responsible for the different fluorescence signatures? If essentially all bacteria show very similar fluorescence signatures, is there any hope to sub-classify bacteria based on WIBS measurements? Several stud-

ies have reported fluorescence signature from bacterial species (Pan et al., 2010;Pan et al., 2007). Some of them should be cited here to check the agreement of fluorescence spectra and WIBS-related "fluorescence type frequencies".

- In line 179-180, the authors state that pollen samples show a "significantly different fluorescence type distribution" (I am ignoring the sizing issue here). This corresponds with published results (Pan et al., 2011;Pohlker et al., 2013). A short statement would be helpful to embed this observation into the existing knowledge.

- In lines 181-188, the authors summarize the "fluorescence response to spore aging". An aging effect is observed for some species. What does this imply for the use of the reported WIBS "reference basis" for ambient measurements? Any chance to visualize this shift in Fig. 3?

Minor and Specific Comments:

- Keywords: I think the use of "aerosol cytometry" is confusing here. There are several studies that apply 'classical' cytometry to bioaerosol analysis (Chen and Li, 2007;Prigione et al., 2004). By using the same term, the different analytical approaches (airborne particles vs. particles suspended in water) may be mixed up.

- Keywords: Probably "WIBS" is another good keyword for this study.

- Line 64: I strongly suggest using the term "intrinsic fluorescence" instead of "fluorescence" here and throughout the entire text. The discrimination between intrinsic and extrinsic fluorescence is a fundamental one and should be very clear from the beginning.

- Line 67: The majority of related studies uses the terms light induced fluorescence (LIF) or ultraviolet light induced fluorescence (UV-LIF)(Toprak and Schnaiter, 2013;Robinson et al., 2013;Healy et al., 2012a;Huffman et al., 2012). Do the authors see any strong reasons to establish another terms, namely "ultraviolet induced fluorescence (UVIF)" in addition? My feeling is that the community should try to avoid using

different terms for the key aspects since this fosters confusion.

- Line 69: "the" is missing in "contribution to airborne carbon pool".

- Line 73: NADH is not defined. Moreover, the biologically correct abbreviation would be NAD(P)H.

- Line 92: In the majority of bioaerosol studies, PBAP refers to primary biological aerosol particles (Despres et al., 2012;Robinson et al., 2013;Perring et al., 2015). Here, it refers to "primary biological airborne particles". I suggest replacing "airborne" by "aerosol".

- Line 101: What is the difference between the "conventional and fluorescent bioaerosol spectra" in this context?

- Line 102: The terms fungi vs. fungal spores should be used more precisely throughout the text: The fungi are cultivated, but the fungal spores are aerosolized.

- Line 108: Replace "20-22C" by "20-22°C".

- Line 112: This sentence refers to "fifteen pure bacterial cultures". In Table 1, 12 cultures are listed, while in Figure 2, 14 cultures are shown. Make sure that these numbers are consistent.

- Line 119: Define "MEA".

- Line 130: Was the WIBS operated in high or low gain mode (Healy et al., 2012a)?

- Line 134: The channel specification here "310-400 nm" does not agree with line 151 "310-420 nm".

- Table 1: It would be very helpful to add information about the scattering of EOD and intensity (e.g., +/- one standard deviation) to get a feeling for the width of these distributions. Moreover, it is not clear where the intensity is derived from. Is it the total fluorescence intensity of the intensity of a specific WIBS channel?

[Figure]

- Figure 2: Replace "fungi" by "fungal spores" in the figure.

- Figure 3: Replace "fungi" by "fungal spores" in the figure. The unit of the x axis is missing.

References:

- Chen, P.-S., and Li, C.-S.: Real-time monitoring for bioaerosols - flow cytometry, Analyst, 132, 14-16, 10.1039/b603611m, 2007.

- Despres, V. R., Huffman, J. A., Burrows, S. M., Hoose, C., Safatov, A. S., Buryak, G., Frohlich-Nowoisky, J., Elbert, W., Andreae, M. O., Poschl, U., and Jaenicke, R.: Primary biological aerosol particles in the atmosphere: a review, Tellus B, 64, 1-58, 10.3402/tellusb.v64i0.15598, 2012.

- Healy, D. A., O'Connor, D. J., Burke, A. M., and Sodeau, J. R.: A laboratory assessment of the Waveband Integrated Bioaerosol Sensor (WIBS-4) using individual samples of pollen and fungal spore material, Atmospheric Environment, 60, 534-543, 10.1016/j.atmosenv.2012.06.052, 2012a.

- Healy, D. A., O'Connor, D. J., and Sodeau, J. R.: Measurement of the particle counting efficiency of the "Waveband Integrated Bioaerosol Sensor" model number 4 (WIBS-4), Journal of Aerosol Science, 47, 94-99, 10.1016/j.jaerosci.2012.01.003, 2012b.

- Hill, S. C., Pinnick, R. G., Niles, S., Fell, N. F., Pan, Y. L., Bottiger, J., Bronk, B. V., Holler, S., and Chang, R. K.: Fluorescence from airborne microparticles: dependence on size, concentration of fluorophores, and illumination intensity, Applied Optics, 40, 3005-3013, 10.1364/ao.40.003005, 2001.

- Hill, S. C., Williamson, C. C., Doughty, D. C., Pan, Y.-L., Santarpia, J. L., and Hill, H. H.: Size-dependent fluorescence of bioaerosols: Mathematical model using fluorescing and absorbing molecules in bacteria, Journal of Quantitative Spectroscopy & Radiative Transfer, 157, 54-70, 10.1016/j.jqsrt.2015.01.011, 2015.

[Figure]

- Huffman, J. A., Sinha, B., Garland, R. M., Snee-Pollmann, A., Gunthe, S. S., Artaxo, P., Martin, S. T., Andreae, M. O., and Poschl, U.: Biological aerosol particle concentrations and size distributions measured in pristine tropical rainforest air during AMAZE-08, Atmos. Chem. Phys., 12, 25181-25236, 2012.

- Pan, Y.-L., Hill, S. C., Pinnick, R. G., Huang, H., Bottiger, J. R., and Chang, R. K.: Fluorescence spectra of atmospheric aerosol particles measured using one or two excitation wavelengths: Comparison of classification schemes employing different emission and scattering results, Optics Express, 18, 12436-12457, 10.1364/oe.18.012436, 2010.

- Pan, Y.-L., Hill, S. C., Pinnick, R. G., House, J. M., Flagan, R. C., and Chang, R. K.: Dual-excitation-wavelength fluorescence spectra and elastic scattering for differentiation of single airborne pollen and fungal particles, Atmospheric Environment, 45, 1555-1563, 10.1016/j.atmosenv.2010.12.042, 2011.

- Pan, Y. L., Eversole, J. D., Kaye, P. H., Foot, V., Pinnick, R. G., Hill, S. C., Mayo, M. W., Bottiger, J., Huston, A., Sivaprakasam, V., and Chang, R. K.: Bio-aerosol fluorescence - Detecting and characterising bio-aerosols via UV light-induced fluorescence spectroscopy, in: Optics of Biological Particles, edited by: Hoekstra, A., Maltsev, V., and Videen, G., NATO Science Series, Human Press / Springer, Dordrecht, 63-163, 2007.

- Perring, A. E., Schwarz, J. P., Baumgardner, D., Hernandez, M. T., Spracklen, D. V., Heald, C. L., Gao, R. S., Kok, G., McMeeking, G. R., McQuaid, J. B., and Fahey, D. W.: Airborne observations of regional variation in fluorescent aerosol across the United States, Journal of Geophysical Research-Atmospheres, 120, 1153-1170, 10.1002/2014jd022495, 2015.

- Pohlker, C., Huffman, J. A., Forster, J. D., and Poschl, U.: Autofluorescence of atmospheric bioaerosols: spectral fingerprints and taxonomic trends of pollen, Atmos. Meas. Tech., 6, 3369-3392, 10.5194/amt-6-3369-2013, 2013.

[Figure]

- Prigione, V., Lingua, G., and Marchisio, V. F.: Development and use of flow cytometry for detection of airborne fungi, Applied and Environmental Microbiology, 70, 1360-1365, 10.1128/aem.70.3.1360-1365.2004, 2004.

- Robinson, N. H., Allan, J. D., Huffman, J. A., Kaye, P. H., Foot, V. E., and Gallagher, M.: Cluster analysis of WIBS single-particle bioaerosol data, Atmos. Meas. Tech., 6, 337-347, 10.5194/amt-6-337-2013, 2013.

- Toprak, E., and Schnaiter, M.: Fluorescent biological aerosol particles measured with the Waveband Integrated Bioaerosol Sensor WIBS-4: laboratory tests combined with a one year field, Atmos. Chem. Phys., 13, 225-243, 2013.

---

## Short Comment (SC1) · 14 Mar 2016

Dear authors,

I have followed your bioaerosol cataloguing work with interest and I agree that a unified approach for UV-LIF calibration and PBAB classification is much needed by the community and the objectives of this study do attempt to address this issue. Referee #2 has already made some useful comments, which I mostly agree with, and I have a few comments which I would also like to add.

Ln 92-95: You state that UV-LIF measurements can only be interpreted using referenced fluorescence emissions. While this is the gold standard we should be striving for, other approaches have been used with success, e.g., hierarchical agglomerative cluster analysis methods have been demonstrated to be useful for interpreting WIBS

UV-LIF datasets collected from forest and mountain field sites (Crawford et al., 2014; Crawford et al., 2015; Crawford et al., 2016; Whitehead et al., 2016).

Ln 130-160: There needs to be a discussion here on how the data has been treated prior to analysis and what QA procedures have been followed e.g., if sampling concentrations were high have weak flashes been removed? What fluorescence threshold has been used to determine if a particle is fluorescent in a given channel? The agreed standard is to use the mean forced trigger value + 3 standard deviations, although other methods have been used. Please clarify this as it is critically important that the same procedure is followed by anyone wishing to interpret WIBS datasets using your results.

Fig. 3: Can you please clarify what the Y axis represents. The Y labels aren't evenly spaced and a number of the Y labels don't correspond to a tick, making it difficult to interpret the figure.

References

Crawford, I., Robinson, N. H., Flynn, M. J., Foot, V. E., Gallagher, M. W., Huffman, J. A., Stanley, W. R., and Kaye, P. H.: Characterisation of bioaerosol emissions from a Colorado pine forest: results from the BEACHON-RoMBAS experiment, Atmos. Chem. Phys., 14, 8559–8578, doi:10.5194/acp-14-8559-2014, 2014.

Crawford, I., Ruske, S., Topping, D. O., and Gallagher, M. W.: Evaluation of hierarchical agglomerative cluster analysis methods for discrimination of primary biological aerosol, Atmos. Meas. Tech., 8, 4979-4991, doi:10.5194/amt-8-4979-2015, 2015.

Crawford, I., Lloyd, G., Herrmann, E., Hoyle, C. R., Bower, K. N., Connolly, P. J., Flynn, M. J., Kaye, P. H., Choularton, T. W., and Gallagher, M. W.: Observations of fluorescent aerosol–cloud interactions in the free troposphere at the High-Altitude Research Station Jungfraujoch, Atmos. Chem. Phys., 16, 2273-2284, doi:10.5194/acp-16-2273-2016, 2016.

Whitehead, J. D., Darbyshire, E., Brito, J., Barbosa, H. M. J., Crawford, I., Stern, R.,

Gallagher, M. W., Kaye, P. H., Allan, J. D., Coe, H., Artaxo, P., and McFiggans, G.: Biogenic cloud nuclei in the Amazon, Atmos. Chem. Phys. Discuss., doi:10.5194/acp-2015-1020, in review, 2016.
* * *

---

## Referee Comment (RC2) · JA Huffman (Referee) · 22 Mar 2016

Manuscript amt-2015-372 submitted by Hernandez et al. presents an overview of laboratory measurements performed using a recently commercialized optical particle counter applied to the detection and characterization of biological aerosol particles. The authors aerosolized representatives from three key classes of bioparticles (i.e. bacteria, fungal spores, and pollen) and present a summary of optical size and fluorescent properties observed. The authors also present a brief comparison of data from two WIBS instruments and a few fungal spores aged for different time periods in order to introduce complexities of applying the WIBS instrument (and other UVIF instruments) to be considered when analyzing field measurements. The manuscript presents a nice introduction of the response of this instrument to these bioparticle types and will be very useful to the UVIF/LIF research community. As someone who has been engaged

in similar lab characterization projects for several years, I hope that the publication of this manuscript can motivate further research in this area. I am confident that AMT is the correct choice for this manuscript and I anticipate that it will be well cited in the near future. In the present form, I suggest some manuscript improvements before publication to improve readability and communication of project scope. After these suggestions have been addressed I happily endorse its publication. (Review comments by Alex Huffman)

General suggestions for improvement: 1. My main overall comment is that I think the scope of the manuscript is somewhat less than may be implied to the reader by the title and portions of the text. While the work is undoubtedly worthwhile, I think some of the wording suggests a broader characterization than was presented here. For example, the title implies to me a relatively comprehensive study that is ready for use as a "library" for the user community. The work is a great step in that direction, but the authors even admit in the text that there are significant challenges to using the WIBS in a standard way for fine-level discrimination. I would suggest changing the title somewhat and also editing a few sections of text to make sure the reader understands the scope of the measurements and conclusions.

a. For example, I would suggest changing the title to something like: "Lab Characterization of Optical Size and Fluorescence Properties of Key Bioaerosol Classes by Wideband Bioaerosol Sensor (or WIBS)". This would clearly communicate the point that the manuscript deals with WIBS data and removes some possible misinterpretations by the reader. Specifically, the clause "composite catalogue" seems too much for the title, "signatures" may be a bit of a stretch for this instrument (as discussed in L207 of the manuscript), and "optical signatures" in contrast to the "fluorescent signatures" to me implies more addition and deeper information than just acquiring and presenting the equivalent optical diameter. b. I would also suggest changing the wording in L97-99, specifically the wording of "systematic compilation of . . .". I think something along the lines of "fluorescent properties and optical size from selective bioparticle

types/species" would be a little clearer. c. The end of the last sentence of the introduction (L98) highlights that the properties "can be reproduced and expanded." Is this sentence meant to imply that these data could be used as a foundation on which other researchers could use to synergize into a "bioaerosol reference basis for a new generation of UVIF instrumentation?" Or do you mean the results are reproducible from instrument-to-instrument? I think clarifying here a bit would help. d. More importantly related to the last comment is that I think that the nice results of the paper actually show that the WIBS instrument, at this stage, is not yet reproducible instrument-to-instrument enough to create a general database useful as a library for unrelated users. For example, Figure 4 suggests strongly to me that there are significant differences between the two instruments utilized for this part of the study. Looking at the category break-down for some of the species you can see that there can be significant differences in the way instrument interpret the same particles. So without some ability to standardize between instruments, the reproducibility may be difficult. e. L91: "cataloguing the optical signatures . . .". Fits in the same set of comments to be somewhat revised. f. Introduction and aspects of the conclusions are somewhat disconnected. For example, the introduction states that one motivation of the work is to acquire and provide a bioaerosol reference basis (L99) and the conclusions state again that the "library" (L249) could be reproduced and expanded. This is certainly a worthwhile goal, but Figure 4 suggests to me that the goal is not yet realized. I would suggest changing the statement in L247, for example, that "this work describes a novel approach for compiling . . ." to something like "an initial approach to compiling . . .". I think this paper is a great first step in this direction, but may motivate continuing work by the authors and others to understand how differences in instruments may improve the reproducibility and ability to draw conclusions from WIBS and other UV-LIF instruments. g. Importantly, however, most of these suggestions amount to a slight re-wording of text and title that shouldn't detract from the importance of the work, but that may help reframe the expectation of readers only able to invest a casual glance. For the community of researchers that may not know much about the WIBS or who have one, but are not as deeply involved with

characterization work, I think this manuscript is a good opportunity to point out some of the important things that should be taken into consideration (i.e. variability between instruments) when interpreting WIBS data.

2. The text is a bit overly concise to convey specific meaning in places. For example: a. L116: The discussion of "direct microscopy showed . . ." is nice, but I would have liked a bit more detail here. Were images of aerosolized bacteria analyzed after/during every experiment, or only once? What kind of microscopy, and at what magnification and resolution? In addition to adding more experimental information here I would suggest adding a few images to the supplement. Consider adding some estimated statistical information (e.g. rougly 1% of bacteria showed . . .; similar to statement made about fungi in L125). b. Also with this concept of aerosolized bacteria, I'm a bit surprised that the process was as simple to collect systematically intact bacterial cells via Collison nebulization and impaction. From (limited) personal experience I have found this to be challenging, and there seems to be reasonably good evidence in the literature that Collison nebulization is violent and can impart significant damage to bacterial cells. Without presenting a summary here, a quick Google scholar search with "Collison nebulizer bacteria damage viability" gives some good examples. I think it would be good to put in one or two references about this topic here and the possible implications it might have on the bacterial part of the study here. c. L125: How was the 1% estimated here and how often? d. L128-129: Similar to the comment above about bacterial aerosolization, what does "notable fractionation of some grains" mean? How was it quantified, and how often was the microscopy performed? I also suggest adding a couple images to the supplement here. e. I would suggest adding some text about the relationship between fluorescence and particle size in the first section of the Results and Discussion. There is a clear increase ($\sim$cubic relationship) between size and observed fluorescence intensity, and so it is not surprising that the pollen show markedly increased fluorescence. The text need not include exhaustive discussion of this topic, but brief mention of the idea (1-2 sentences) would help round out the discussion. There are several references by Pan/Pinnick/Hill et al. that discuss this, as

well as several others. f. L188 suggests that "B Channel" fluorescence increased with spore aging, but doesn't give any suggestion why. Can the authors suggest a possible reason for this?

3. Was the asymmetry factor (AF) measured as a part of this study? I understand that it is an unreliable tool at present, but I think it would be worthwhile to actively state something in the text on this topic, even if it is to say that the AF is not reproducible and will need to be investigated further, etc. Many WIBS users will look to this manuscript for guidance when interpreting bioaerosol data, and a brief statement here would be very useful.

4. The nomenclature suggested in L151 feels subtly different than what Perring et al. 2014 and others have recently used for WIBS data. In the Perring et al. paper (i.e. footnote under Table 1) there was a clear mention of the difference between "Channel A", meaning any particle fluorescing in the FL1 channel, and "Type A" meaning a particle that fluoresces in Channel 1 / FL1 but NOT in either of the other two channels. This is maybe a subtle difference, but probably a useful one for the WIBS community to become/stay consistent with. I personally like the term e.g. FL1 to refer to the channel and the e.g. Type A to refer to the active statement of the combination of channels exhibiting fluorescence for a given particle. However, I'm also happy with the terminology utilized by Perring et. al, 2014, which I think is a bit clearer than using "Type A" to mean any particle fluorescing in A (even if also in B and C) as implied in L151 here.

5. I suggest expanding the caption from Table 1 to include details about what the numbers mean. For example, it appears that the fluorescence type frequency is normalized to a sum of 100, but this is not explicitly mentioned. There are no units on EOD or Intensity, and 'samples' is not clearly defined as 'observed particles' here. I would also suggest adding a standard deviation for the EOD and intensity measurements here. The text makes mention of a range of observed properties within a particle type, and this range should be reported in some form here.

6. L136-138: This was a confusing sentence to me. What does the 2.0 and 2.8 um mean? This suggests to me that a two-point size calibration was done, but I doubt I'm interpreting this correctly. Can you say briefly, but specifically more detail about the size calibrations performed (by PSLs over X-Y range, etc.)

7. L182-186: This statement is a bit confusing to me. Looking at Figure 2, there are some differences in the young/old spores (e.g. ratio of A and AB) as mentioned. The cladosporium difference (considered indistinguishable) doesn't look much difference than the phoma herbarium difference (considered a significant shift). I would consider defining or tightening up how these statements were made. Along these lines, I would suggest considering to pull out the aging effect of the spores into a separate plot. It is really hard to see this difference in the current Figure 2, and it would show the point clearly by putting only the young/old together.

8. One conclusion (L252-253) states that "...primary physiologies can be unambiguously differentiated from each other ...". This is true, with respect to Figure 3, but the only clearly discernable difference between most of the fungal spores and bacteria is the size. The fluorescence properties (i.e. breakdown of categories, FL1 intensity) are generally within range of one another. So assuming the bacteria are aerosolized individually in the lab, this may be possible, but I think an important conclusion that is nicely shown here is that extrapolating this technique to the atmosphere may make differentiation between bacteria and spores exceedingly difficult. Would it make the differentiation easier to see in Figure 2 to use FL3 as the color scale rather than FL1?

Minor / specific comments: • I suggest adding section numbering to help organize and guide the reader. • I'm not familiar with the term UVIF, but have seen UV-LIF for laser and light-induced fluorescence. Does UVIF have a standard history in some area of scientific literature? If so I suggest adding a reference in the second paragraph to this history, simply because it may be that the atmospheric measurement audience may be more familiar with the other. If not, I would suggest using the UV-LIF terminology. • Was the WIBS inside the chamber? I had understood that previously,

but the text implied otherwise. • L70: Indeed, fluorescence-based instrumentation has been utilized frequently in recent years as a part of many measurement studies. I would suggest adding one by Pan et al. (e.g. JGR, 2007) and also one by Huffman et al. (e.g. ACP, 2013 or ACP, 2012) to round out some of the groups that have published a number of studies in this area. • L103: chamber . . . "is" cubic (instead of was) ? • L136: space between 635 and "nm" • L143: How likely was re-suspension of large particles in a subsequent test? Was physical cleaning ever performed in the case that UV light, ozone, ethanol vapor did not physically wash out large particles? • L192: monodispersed is misspelled • L216: Cite Pohlker et al., AMT, 2013 regarding fluorescence spectra of pollen. • L254: The text brings up the mention of "deeper cluster analysis," which I agree is beyond the necessary scope of this text. I still think at this mention it would be useful to cite the Manchester team who has been working on this area, e.g. Robinson et al. 2013 or Crawford et al. 2014 or 2015. • Is there a reason why Table 1 lists 1-12, then skips to 28-56 before returning to 15-27? • Were two instrument "calibrated" with any of the same particles at same time (with all four channels)? I realize this is a difficult task and one of on-going research efforts. However, if no standardization of signals was attempted, it is hard to know how much to trust Figure 4, or not. E.g. L221-222 mentions gain settings, etc. Were these standardized? • Figures 2, 4: The bar width scale is confusing. Why is the scale approximate? • Figure 3 has no units • L52: remove "some intact" from sentence • Add UV-LIF (or equivalent) as keyword • L51: remove capitalization from ultraviolet • L86: Statement about "short stability window" needs to explicitly say time somewhere in the sentence to be less ambiguous. Also, this statement needs a reference of some kind, even if to say "personal experience. • L96 and 132: I'm confused by the addition of the word "portable." Aren't all WIBS instruments portable? Is the "portable variant" the Introscope (?) version? If so, I think this should be mentioned in the methods section to highlight the differences between the WIBS and whatever the portable "variant" is. • L104 space between 1.5 and W • L107, L108, L120 space between number and unit: m, C, C • L136-138

sentence is long and should be shortened and simplified • L167, L179: avoid using "distinctly clustered" to avoid confusion with cluster analysis • L179: Is the correct term "pollens" here? I would suggest species or pollen types to disambiguate. • L194: comma before however • L208: Add citation to Perring et al. and Manchester clustering efforts (this is already being done) • L211: Cite Pan et al. here. Also, briefly explain what you mean by "statistical treatment". If you mean clustering, then state specifically and cite. If something else, I would add another few descriptive words. • L219-220: Cite Pohlker et al. 2013

---

## Author Comment (AC1) · 27 Apr 2016

We thank Dr. Crawford for his helpful comments and criticisms and address them below in the order they appeared.

Ln 92-95 - We agree with this reviewer's opinion about the absolute nature of our statement; indeed UV-LIF data can be interpreted with utility in many different credible ways, notably including hierarchical cluster analyses. By no means did the authors intend to suggest the approach described here was the "gold" standard for UV-LIF optical particle recognition; we mean to present a systematic approach which, from an aerobiology perspective, can be replicated (or expanded) in a referential "library" paradigm. We have amended the manuscript to reflect (and reference) the fact that different analytical approaches — including cluster analysis— are valid, and can

be successfully used to characterize aerosol particle challenges to UV-LIF instruments both in the environment and in the laboratory. In this specific regard, we have expanded the citations suggested by this reviewer in the context of this investigation (and its discussion). The authors would like to point out that the process of this pure-culture library challenge manifests in a defacto cluster analyses which resolved, as the title suggests, the physiology of three major bioaerosol classes using UV-LIF as configured in a WIBS.

Ln 130-160 - With regard to discussion about the fluorescence thresholds used to qualify these data for analyses, we have amended the methods section of the manuscript in accordance with this reviewer's suggestion. As is customary in this sub-discipline, we did follow the practice of using a (mean) forced trigger value + 3 Standard Deviations.

Fig. 3 - Dr. Crawford (and another reviewer) asks for clarification and amendment to the Y axis of figure three. We apologize for this graphical oversight and have corrected the symmetry and labels on the ordinate axis and expanded this figure's caption.

---

## Author Comment (AC2) · 27 Apr 2016

We thank this anonymous reviewer for his/her helpful comments and criticisms. We address them below in the order they appear.

General Comments

We apologize for any oversights and editorial inconsistencies between the tables, text and figures in regard to the formal Latin names of the cultures used in this study. We have reviewed the Latin spellings for all the source cultures used in this investigation, and cross-checked them against each other. Those that were misspelled were corrected.

Ln 97-99. The authors have chosen to use the term "systematic" in a generic context,

as it refers to the method described here. We realize however, this can be confused with microbial systematics in the context of taxonomy. In this context we make no claims nor do we intend to present any inference as such. Thus we have removed the word "systematic" in this context.

Justification and citations for the selection of microorganisms used in this study: The microbial inventory of the atmosphere remains a topic of intense study, which has recently been accelerated by high throughput DNA sequencing. While genotypic characterization is advancing our understanding of the relative abundance atmospheric microbes, classical culture and microscopy still represent the majority of aerobiology investigations published to date. Optical properties, whether fluorescent or not, are phenotypic properties. Thus, to demonstrate this library approach, we chose a subset of pure-cultures of bacteria, fungi and pollens which have been (repeatedly) recovered and identified from aerosols in both indoor and outdoor environments by culturing a microscopy. Of the 14 bacterial cultures used here, all but two cultures are medically relevant to public health or bioterrorism, and eight cultures have (commonly) been used as bioaerosol models to their persistence under different atmospheric / disinfection conditions (e.g. UV). The three bacterial cultures used here that have not been previously recovered from the atmospheric environment, or otherwise used in prior bioaerosol studies (Thiobacillus sp., Vibrio sp. and Enterobacter sp.) were simply chosen to broaden the range of bacterial phenotypes used for these fluorescence challenges such that all the major bacterial physiologies were represented: Gram Positive, Gram Negative, bacterial spores, cocci, bacilli, and vibrio (filamentous bacteria were purposely not included).

With respect to the fungi and pollens chosen, the same logic applied in choosing which cultures to use for this demonstrative library (challenge): all members of the 26 fungal cultures used and all 13 pollen grain stocks have been recovered from different atmospheric environments. While certainly not exhaustive, these cultures cover a broad range of fungal and pollen physiologies commonly recovered from the atmospheric

environment (both indoors and out), and reported on a phenotypic basis. This discussion has been summarized and supporting references added to the manuscript as this reviewer has requested.

Response to reviewer's comments regarding cultivation history: The authors are in agreement that this manuscript (re)states that "cultivation history" can be a factor in observing fluorescence properties of airborne microorganism with UVIF; however, we present this statement through a carefully qualified citation [Saari et al, 2014]. As such, we believe that this statement is appropriately cited and presented. Because of the significant length it would add the manuscript, the authors purposely chose to minimize the (routine) level of microbial culturing detail this reviewer is asking for—since it can be easily accessed from their source collections (ATCC and DSMZ) or classic microbiology lab manuals. We believe that providing culture details, beyond where the cells were harvested in their growth cycle (e.g. early stationary phase) and their immediate preparation prior to aerosolization, is considered so routine (and protocols so easily attained) that this information should at most be included in supplementary information (if at all). We believe the length that generic culturing protocol material would add to the manuscript would distract from the main point of the work, and respectfully decline to do so with exception to separate supplementary files. We have however, expanded the manuscript to include more culturing details with respect to harvesting (cultivation time) and immediate cell preparation. In response to the reviewer's request, we have added ATCC and DSMZ strain designations to table 1, where appropriate.

Response to reviewer's comment with respect to determining "spore bearing physiology": Each culture was observed with phase contrast microscopy and classical spore staining to ensure that spores we present and dominant. These classic methods were cited in an expanded version of this manuscript. Response to reviewer's comment with respect to determining "spore bearing physiology: The pollens were collected directly from their sources at the Botanic Garden cited, and stored dry (between 20-40% RH). Unlike bacteria and fungi however, pollen granules are not cultured, but acquired from

their botanical source; pollen grains do not have classical microbial growth cycles is not practically possible to determine their age.

Response to reviewer's comment with respect to referential basis for UVIF: We respect to this anonymous reviewer's opinion that THIS particular cohort of fluorescence data can serve as reference basis for new generations of UVIF instrumentation – we agree this statement is too broad. We also believe this reviewer is simply pointing out a semantic issue around the word "can". We thus amend this statement in an updated version of the manuscript to generically state that this type of referential approach may (colloquial equivalent to "can") add value to the bioaerosol characterization field. We amend the text as the reviewer requests to explicitly present this work as a reference method (not about the absolute value of these reference data).

Ln 128-129 and Ln 175-180. The authors acknowledge(d) that many different types of pollen grains often fractionate in aerosols: this fractionation is well documented to naturally occur in the atmosphere (as well as in the laboratory), and references have been added to support an expanded discussion which include this bioaerosol behavior. Because of this phenomena, the EOD distributions for pollen grains are much wider than their fungal or bacterial counterparts, and the means of these distributions, as acknowledged, are less than unadulterated pollen grains, which often have true optical diameters larger than 10 um. We expand our discussion to better present the fractionation which was observed in this study, but are reticent to remove our reporting of pollen grain fluorescence for the reason that their fluorescence type distributions are so markedly different than fungal spores or bacterial cells – this is a noteworthy finding and the authors believe this juxtaposition is worth presenting in this methodological context.

Ln 116. The type of microscopy used was classic phase contrast and fluorescence microscopy (x1000), which the authors believe was appropriately referenced in the original version of the manuscript (citing bioaerosol studies which explicitly include the survivability of many of these same bacterial cultures through nebulization). Using

modern microscope equipment, those skilled in this art reliably resolve submicron dimensions of bacterial cells using reticules calibrated on this scale (however, this is a direct optical measurement, and not an Equivalent Optical Diameter (EOD)). Based on previous studies in this and other bioaerosol laboratories (citations added), with many of the same bacterial models used here, the authors acknowledge that some microbes, particularly Gram negative bacteria harvested in their log growth phase, can experience significant viability losses after being refluxed in Collison nebulizers even after 10 minutes. To this end, we purposely harvested bacteria in early stationary phase and held nebulization times to 2 minutes to minimize variance with viability – this was a conscious choice in compiling and executing this method. Thus, we did not measure culturability as it was not central to this investigation. The revised version of this manuscript, includes an expanded of detail of microscopy and aerosolization procedures.

Response to reviewer's comment with respect to previous UVIF studies including similar culures: We appreciate this reviewer's perspective and his/her request to juxtapose the results reported here, to those previously reported from other studies, which isolated UVIF response of similar, if not identical cultures (as catalogued by ATCC or DSMZ). Since this is an archival journal of Measurement Techniques, our point was to report the method in a concise manner as this reviewer suggests, which a respectable cohort of demonstrative data. As requested, in the revised version of this manuscript, we juxtapose what subset of pure culture bioaerosol (fluorescence) data from other studies can be legitimately and practically compared, including those reported by Hill, Healy, Pan, Pohlker and their coworkers (circa 2001-2014), although many of these previous studies included substantial focus on asymmetry factors, analyzed significantly less bioaerosol (102-3 particles as compared to > 104) and/or did not specifically provide for equilibrium between the chamber humidity and bioaerosol.

Ln 162-176 and Ln 179-180. The authors agree with the reviewer's suggestion that we state how these results show clear differences in fluorescence (distribution) patterns

between pollen(s) and pure cultures of fungi (spores) and bacteria. As suggested above, we (will) acknowledge (and cite) that Pan, Pohlker and their coworkers also observed differences in UVIF signatures between certain pollens and other cultures of airborne microbes – although those cultures and aerosol conditions were different than those reported here.

Ln 169-170. The reviewer reiterates the authors' statement referring to the fact that the only culture of sporulated bacteria aerosolized in this study (B. subtilis), was markedly different from the other bacteria cultures aerosolized in terms of its fluorescence distribution. This is a straightforward observation. In response to the prompt by this reviewer, the authors make no claims or inferences in the current (or future) form of this manuscript regarding the ability of this UVIF equipment configuration to otherwise sub-classify bacterial cultures. Since only one type of bacterial spore was used, we are reticent to expand discussion beyond this simple observation.

Ln 181-188. The reviewer reiterates the authors' statement referring to the fact that of four different fungal cultures observed, modest differences in fluorescence distributions (A shifting to AB) were noted between otherwise young (28 d) and aged (180 d) fungal spores: P. crysogenum, P. herbarum, and C. cladosporioides. In response to the prompt by this reviewer, the authors make no claims or inferences in the current (or future) form of this manuscript regarding the ability of this UVIF equipment configuration to otherwise sub-classify fungal spores based on age. While this does, however, beg efforts to specifically study the effects of aging on fluorescence distribution (and intensity); we are reticent to expand discussion beyond this simple observation.

We have addressed the following "minor" comments as requested by this anonymous reviewer: Keywords:

The term "aerosol cytometry" has been dropped from the text to avoid any potential confusion. Keywords: WIBS has been added to the keyword list.

Ln 64: intrinsic fluorescence will replace "fluorescence" where appropriate in context

throughout the text.

Ln 67. UV-LIF will be used as the acronym to describe all instance of ultraviolet light induced fluorescence and any variant thereof.

Ln 73. The first instance of Nicotinamide Adenine Dinucleotide Hydrogenase will be introduced and thereafter referred to by acronym, NADH and its phosphorylated derivative NADPH.

Ln 92. The first instance of Primary Biological Aerosol Particle will be introduced as consistent with the recent literature and thereafter referred to by acronym- PBAP.

Ln 101. Conventional (light) will be delineated from fluorescent spectra.

Ln 102. The culture of fungi (pl.) will be delineated from fungal spores throughout the text.

Ln 108. In all instances, the reporting of temperature within ranges in the Centigrade scale will be reported with the degree symbol preceding a capital C (e.g. $X^o$ C).

Ln 119. MEA is malt extract agar, which is commonly used for the cultivation of fungi. Its first instance will be introduced, and thereafter referred to as MEA.

Ln 130. The WIBS was operated in low gain mode; this operational setting will be added to the methods section. Ln 134. The typographical error for the channel specification in the range between 310-400 nm has been corrected throughout the text.

Table 1. Standard deviations for EOD have been added to Table 1.

Figures 2 and 3. Fluorescence intensity in Figure 3 is associated with the range between 310-400 nm "Fungi" has been replaced with "Fungal Spores" in Figures 2 and 3.
* * *

---

## Author Comment (AC3) · 27 Apr 2016

We thank Dr. Huffman for his helpful comments and criticisms and address them below in the order they appear. The title has been changed to reflect that the work was limited to chamber assessments on a meso-scale (1m3): Chamber based catalogues of optical and fluorescence properties can distinguish bioaerosol classes

Ln 97-99. The authors have chosen to use the term "systematic" in a generic context, as it refers to the method described here. We realize however, this can be confused with microbial systematics in the context of taxonomy. In this context we make no claims nor do we intend to present any inference as such. Thus we have removed the word "systematic" in this context. (Ln 98) We mean to present the perspective of generalized, chamber-based bioaerosol challenge method for UV-LIF, which can be

applied to generate this quality of optical and fluorescence response catalogues. Ln 99, 247 and Figure 4. The authors agree with the reviewer's perspective of figure 4; thus, the discussion has been expanded to highlight this operational example and the fact that different WIBS instrumentation, while configured the same, can generate (somewhat) different aerobiological catalogues as judged by fluorescence distributions of pure culture microbes aerosolized in chambers under the same conditions.

Ln 116. The type of microscopy used was classic phase contrast and fluorescence microscopy (x1000), which the authors believe was appropriately referenced in the original version of the manuscript (citing bioaerosol studies which explicitly include the survivability of many of these same bacterial cultures through nebulization). Using modern microscope equipment, those skilled in this art reliably resolve submicron dimensions of bacterial cells using reticules calibrated on this scale (however, this is a direct optical measurement, and not an Equivalent Optical Diameter (EOD)). Based on previous studies in this and other bioaerosol laboratories (citations added), with many of the same bacterial models used here, the authors acknowledge that some microbes, particularly Gram negative bacteria harvested in their log growth phase, can experience significant viability losses after being refluxed in Collison nebulizers even after 10 minutes. To this end, we purposely harvested bacteria in early stationary phase and held nebulization times to 2 minutes to minimize variance with viability – this was a conscious choice in compiling and executing this method. We did not measure culturability as it was not central to this investigation. The revised version of this manuscript, includes an expanded of detail of (these routine) microscopy and aerosolization procedures.

Ln 128-129. The authors acknowledge(d) that many different types of pollen grains often fractionate in aerosols: this fractionation is well documented to naturally occur in the atmosphere (as well as in the laboratory), and references have been added to support an expanded discussion which include this bioaerosol behavior. Because of this phenomena, the EOD distributions for pollen grains are much wider than their fungal or

bacterial counterparts, and the means of these distributions, as acknowledged, are less than unadulterated pollen grains, which often have true optical diameters larger than 10 um. Direct microscopy was performed on subsets of experiments with each different species aerosolized, and the influence of pollen fractionation was evident; however, pollen image analysis was not central to this study. We expand our discussion to better present the fractionation which was observed in this study. Results and Discussion. As requested, in the revised version of this manuscript, we juxtapose what subset of pure culture bioaerosol (fluorescence) data from other studies can be legitimately and practically compared, including those reported by Hill, Healy, Pan, Pohlker and their coworkers (circa 2001-2014), although many of these previous studies included substantial focus on asymmetry factors, analyzed significantly less bioaerosol (102-3 particles as compared to > 104) and/or did not specifically provide for equilibrium between the chamber humidity and bioaerosol. In some cases (figure 3), fluorescence intensity was notability higher with pollen grains regardless of fractionation.

Ln 181-188. Of four different fungal cultures observed, modest differences in fluorescence distributions (A shifting to AB) were noted between otherwise young (28 d) and aged (180 d) fungal spores: P. crysogenum, P. herbarum, and C. cladosporioides. In response to the prompt by this and other reviewers, the authors make no claims or inferences in the current (or future) form of this manuscript regarding the ability of this UV-LIF equipment configuration to otherwise sub-classify fungal spores based on age (in any channel). While these data beg efforts to specifically (and extensively) study the effects of aging on fluorescence distribution (and intensity), we are cautious to expand discussion beyond this simple observation.

Response to reviewer's comment with respect to asymmetry factor: Because of its variance, we purposely avoided including asymmetry factors in this cataloging demonstration study – in fact we did not collect this data during these experiments. We cannot formulate a consensus about how to incorporate a generalized statement about asymmetry factors given we did not collect these data.

Line 151. We have cited Perring et al, 2015 as generalized guidance for cataloging fluorescence distribution response(s) of the pure cultures aerosolized here, and report our results in this format as such. As requested, we have expanded our explanation of Perring reporting format (and legends) in a revised version of this manuscript.

Table 1. We agree with this reviewer's observations and will explicitly present the fluorescence distributions on a percent (%) basis. Standard deviations for EOD have been added to Table 1.

Line 136-138. An expanded and detailed discussion of the two-point (PSL) instrument calibration has been added to a revised version of the manuscript.

Line 182 – 186. We acknowledge the density of figure 2; however, we have considered pulling out young (28d) and aged (180d) comparisons in a separate plot. This would unfortunately add significant graphical redundancy to the manuscript. We have however, added detail in the discussion of what differences have been noted between the fluorescence distributions of three cultures young and ages spores (see comment above Ln 181-188) in a revised version of the manuscript.

Ln 252-253. As a point of clarification in this context, primary physiology refers to that defining the major bioaerosol classes based on their differences in structural phenotype: virus, bacteria, fungal spores and pollen grains. Figure 2 juxtaposes fluorescence distribution and EOD of these major classes (not including virus) where distribution is represented across (all) seven possible fluorescence combinations for a given (pure culture) population.

We have addressed the following "minor" comments as requested by this reviewer:

Ln 67. UV-LIF will be used as the acronym to describe all instance of ultraviolet light induced fluorescence and any variant thereof.

Request and Clarification: We don't understand this reviewer's request for "standard" history. Given this is a methods presentation, the authors believe the motivation and

historical context are adequate.

Request and Clarification: The WIBS was not "inside" the chamber, and the methods section has been amended to better reflect the apparatus configuration.

Ln 103. The tense describing the state of the chamber was changed from was to is (present).

Ln 143. Resuspension was unlikely given the evacuation and cleaning protocol and confirmed at the beginning of each test by particle monitoring. As judged by particle monitoring, evacuation and cleaning never failed to remove large particles.

Ln 192. Spelling of monodispersed corrected.

Ln 254. Cluster analyses have been acknowledged and citations added in accordance with reviewer 1.

Table 1. The numbers correspond to cultures as grouped by their major physiologic class, but have been removed for clarity.

Request and Clarification: Both instruments were calibrated with PSLs obtained from the same source, but they were not calibrated at the same time.

L221. Gain settings were not "standardized" between instruments, but both instruments were operated in their lowest gain range. We have added this statement to the methods section of a revised manuscript.

Figure 2. The legend for this figure includes a horizontal bar, which represents the scale for EOD. It is approximate simply because it cannot be directly calibrated with pixels in the graphical program used to create it (Igor v 6.0).

Figure 3. Like Dr. Crawford, Dr. Huffman asked for clarification and amendment to figure three. We apologize for this graphical oversight and have corrected the symmetry and labels on the ordinate axis and expanded this figure's caption.

Ln 52. The words "some intact" have been removed from this sentence.

Ln 86. The statement about the "short stability window" has been expanding to include a time frame on the order of "days".

Ln 92 and Ln 132. The term "portable" has been removed from describing the WIBS, and the serial numbers for the WIBS used in this study, are included in the methods section of a revised manuscript

Ln, 104, 107, 108 and 120. Spaces removed.

Ln 136-138. Sentences shortened.

Ln 167. The term "clustered" was changed to "related assemblages"

Ln 179. Pollens is the correct generic plural of pollen, as referred to different physiological types or species.

Ln 208, 211, and 219. The citations requested have been added.